# Effectiveness of a preoperative orlistat-based weight management plan and its impact on the results of one-anastomosis gastric bypass: A retrospective study

**Hung-Chieh Lo**[1,2,3]*, **Shih-Chang Hsu**[4,5]

**1** Division of Trauma and Emergency Surgery, Department of Surgery, Wan Fang Hospital, Taipei Medical University, Taipei, Taiwan, **2** Department of Surgery, School of Medicine, College of Medicine, Taipei Medical University, Taipei, Taiwan, **3** TMU Research Center for Digestive Medicine, Taipei Medical University, Taipei, Taiwan, **4** Department of Emergency, School of Medicine, College of Medicine, Taipei Medical University, Taipei, Taiwan, **5** Emergency Department, Wan Fang Hospital, Taipei Medical University, Taipei, Taiwan

\* carfishcat@yahoo.com.tw

**Data Availability Statement:** All relevant data are within the paper and its Supporting Information files.

## Abstract

### Introduction

The aim was to first investigate the efficacy of a preoperative weight management program centered on orlistat, which is mechanistically similar to gastrointestinal bypass procedures in that it restricts dietary fat absorption, and then assess its impact on the results of one-anastomosis gastric bypass (OAGB).

### Materials and methods

We retrospectively reviewed the clinical data of consecutive patients aged 20–65 years with a body mass index (BMI) $\geq$ 42.5 kg/m$^2$ who underwent primary OAGB from 2014 to 2020. Eligible patients who adhered to a 10–14 day orlistat regimen as part of a 4–6-week diet/life-style modification plan preceding surgery were stratified into weight reduction (Group 1) and weight gain (Group 2) groups post treatment. The correlation between pre- and postoperative weight loss and perioperative outcomes was assessed.

### Results

Of 62 eligible patients, 55 met the inclusion criteria and complied with treatment; 35 (64%) patients in Group 1 lost a median of 2.0 kg, and Group 2 had a median weight gain of 2.9 kg. Group 1 had a significantly higher initial BMI (48.9 kg/m$^2$ vs. 44.6 kg/m$^2$; p = 0.003), more females (54% vs. 25%) and a shorter operation time than Group 2 (107 min vs. 140 min; p = 0.109). There was no difference in the incidence of 30-day complications. Weight loss did not differ between the groups at 24 months.

**Funding:** The author(s) received no specific funding for this work.

**Competing interests:** The authors have declared that no competing interests exist.

## Conclusion

Effective weight control through an orlistat-containing regimen benefitted two-thirds of patients who underwent OAGB; however, further weight loss was not observed at 2 years post-surgery.

## Introduction

Bariatric and metabolic surgeries are widely accepted as the treatment of choice for obesity and its associated comorbidities. While natural weight gain is not uncommon in the waiting period and can lead to continued health deterioration [1], body weight per se is a well-known risk factor that can be altered via comprehensive preoperative planning [2]. Evidence indicates that modest preoperative weight loss has beneficial health effects [3] and reduces the risk of complications [4, 5]. In the relevant guidelines, proper preoperative intervention has been suggested to improve technical aspects [6]. Nevertheless, no consensus has been reached regarding the indications for intervention or standardized approaches. It has been estimated that more than 70% of surgical candidates fail to achieve meaningful weight loss via lifestyle modifications alone [1]; instead, regimens that combine a low-calorie diet with relevant medications or endoscopic-based interventions are considered more effective [7]. However, the implementation of such regimens requires access to local resources and feasible applicability. Due to the lack of standardized guidelines, the preoperative determination of body mass index (BMI) cutoff points is based on individual assessment and greatly influenced by the specific characteristics of the study populations. For example, a systematic review revealed substantial variation in the BMI cutoff point for initiating weight management programs, as the baseline BMI ranged from 43.5 to 58.4 kg/m$^2$ [4]. In Sweden, the average BMI at the time of surgical planning was 42 kg/m$^2$, and a 2- to 3-week low-calorie diet regimen was routinely implemented, regardless of the preoperative BMI [8]. Studies have shown that in terms of BMI, Asians tend to have a higher body fat percentage than matched Caucasians and a higher prevalence of central adiposity than Europeans [9, 10]. We have also observed in our own prior experience that the success of the procedure was hindered in patients with a relatively low BMI (ranging from 40.8 to 48.1 kg/m$^2$) and a voluminous liver [11]; as the goals are to ensure the safety of our patients and the success of the procedure, we decided to implement a prudent preoperative weight management program for patients with a lower BMI than that specified in Western reports [12]. In terms of procedure selection, one-anastomosis gastric bypass (OAGB) has been increasingly performed worldwide [13], is acknowledged to have a low-risk profile and is considered an appropriate procedure for patients with a BMI >50 kg/m$^2$ [2, 13]. We tend to prefer this kind of surgery considering the level of evidence for those with severe obesity. To facilitate procedure progression and increase safety, a pragmatic 4- to 6-week intervention program including 10–14 days of orlistat treatment (Xenical®; Roche Products, Basel, Switzerland) is routinely recommended for patients with BMI ≥42.5 kg/m$^2$ because orlistat was the only available weight management medication with reported efficacy and licensed by the local Food and Drug Administration across the study period [14]. Patients were instructed to self-administer orlistat to increase adherence. Limiting dietary fat intake is not only a dietary requirement after surgery but also a prerequisite to avoiding the side effects of orlistat, and the regimen also provides patients with an opportunity to adapt to a low-fat diet in advance. Initially, we found interindividual variations in the response to such a program. It is hypothesized that patients with better adherence and a favorable response may have excellent surgical outcomes.

The current study was undertaken to review the effectiveness and appropriateness of a routine weight management protocol that centered on the mechanism of orlistat over a 7-year period and to assess its impact on medium-term weight loss and 30-day surgical outcomes after OAGB.

## Materials and methods

We retrospectively reviewed a prospectively maintained database of consecutive patients who underwent bariatric surgery from 2014 to 2020 by a single surgeon at a university-affiliated hospital. Local institutional review board approval was obtained. All procedures in this study were performed in accordance with the ethical standards in the 1964 Declaration of Helsinki and its later amendments. The research project was approved by the Taipei Medical University-Joint Institutional Review Board (No.: N202109040). Informed consent was waived because no case data were disclosed. The study began on September 23, 2021, and subject data were collected from the start of the study to October 21, 2021. All patient data are coded to ensure the privacy of the subjects and the confidentiality of the data.

Patients aged 20–65 years with a BMI $\geq$ 42.5 kg/m$^2$ who underwent primary OAGB were included. Patients who underwent revision procedures, took medication that could interact with orlistat or had a history of chronic diarrhea were excluded. Weight gain was discouraged, but patients were not excluded from surgery on this basis. Routine preoperative screening included a psychological evaluation that specifically addressed the presence of eating disorders, a complete metabolic panel, electrocardiography, chest radiography, and esophagogastroduodenoscopy. To increase adherence and accommodate cost-based considerations across the study period, a consistent approach was advised, including orlistat (120 mg) three times daily immediately after meals in the 10–14 days preceding surgery as part of a comprehensive 4- to 6-week self-management plan, which consisted of lifestyle modification (increased physical activity and a low-fat, low-carbohydrate, high-protein diet) and structured education on how to avoid adverse gastrointestinal events and maintain medication compliance. The total out-of-pocket cost is US$25–35, and the acceptance or rejection of the regimen was at the patient's discretion. Written informed consent was obtained from patients who acknowledged the aforementioned information and were willing to pay for their own treatment. Procedure selection was conducted through a shared decision-making process, except OAGB was preferably provided to those with a BMI $\geq$ 50 kg/m$^2$. The current general selection criteria correspond with the previous selection guidelines in that grade C or D gastroesophageal reflux disease (GERD) or Barrett's esophagus are contraindications for the index procedure in both guidelines [15]. In the specified surgery, a long, narrow gastric pouch was safely and effectively created by stapling linearly, starting at crow's foot and extending just lateral to the angle of His alongside a Fr. 32 calibration tube. Due to its reported safety and efficacy, a biliopancreatic limb with a fixed length of 200 cm was used [16]. Patients were hospitalized the day before the operation. The responsible staff recorded the patient's weight control methods utilizing a comprehensive questionnaire that inquiries about dietary modifications and gastrointestinal adverse events. Additionally, we specifically assessed the frequency and timing of daily medication use to ensure the accuracy and reliability of self-reported outcomes and to assess medication adherence. Preoperative weight change was calculated as the difference between the weight obtained at the admission date and at the last clinic visit using the same digital scale and described as the percentage of total weight loss (%TWL) and the percentage of excessive weight loss (%EWL). A BMI of 23.5 kg/m$^2$ was used when individuals from Taiwanese [17] and Asia-Pacific ethnic backgrounds were included in the study population [18]. Patients were stratified into 2 groups: those who lost any amount of weight (Group 1) and those who gained

any weight (Group 2). Postoperatively, all patients received the same health education at scheduled appointments at 1 week and 1, 4, and 12 months postoperatively and then annually thereafter. The outcomes assessed were preoperative weight control, operation time, 30-day complications and weight loss trajectory up to 3 years postoperatively.

## Statistical analysis

Continuous variables are expressed as medians and interquartile ranges. Categorical variables are summarized by using descriptive statistics with frequencies (percentages) or counts. The Mann–Whitney U test and Fisher's exact test were utilized for comparisons between groups. Statistical tests were two-sided, and a p value $< 0.05$ was considered statistically significant using R 3.6.1 software (R Foundation for Statistical Computing, Vienna, Austria).

## Results

From 2014 to 2020, a total of sixty-two patients met the inclusion criteria. Of these patients, one patient declined to participate. Six patients withdrew from the study early: one missed medication, three experienced a weight plateau, and two had gastrointestinal adverse events. For the remaining fifty-five eligible patients who fully adhered to treatment, 35 (64%) presented with weight loss of varying degrees (Group 1), while the remaining 20 patients had weight gain (Group 2).

The clinical characteristics are outlined in Table 1. Group 1 had younger (33.0 years (44.5–27.5) vs. 37.5 years (48.0–33.0); p = 0.180) patients, included more females (54% vs. 25%; p = 0.068), and had a significantly higher BMI at the last clinic visit (48.9 kg/m$^2$ (52.7–45.8) vs. 44.6 (46.8–42.8); p = 0.003) than Group 2. No significant differences were found between the groups with respect to BW and BMI at admission or the prevalence of comorbidities. There was a median weight reduction of 2.0 kg in Group 1, in contrast to the 2.9 kg weight gain in Group 2. The median preoperative %TWL and %EWL were 1.7% and 3.1% in Group 1 vs. 2.2% and 4.9% weight gain in Group 2, respectively.

As depicted in Table 2, the median operation time was reduced in Group 1 (107.0 min (155.5–80.0) vs. 140.0 min (169.5–100.5); p = 0.109), whereas there was no difference in length of stay or 30-day complications.

Follow-up data are available for 78% vs. 75%, 62% vs. 58, and 59% vs. 22% of patients in Group 1 and Group 2 at 12, 24, and 36 m, respectively. As shown in Fig 1A, a significantly greater %EWL was found in Group 2 than in Group 1 at 1 week (12.8% vs. 9.7%; p <0.05), 1 month (22.4% vs. 19.3%; p = 0.031), and 4 months (44.5% vs. 39.8%; p = 0.045), and a larger % TWL was observed at 1 week (6.1% vs. 4.8%; p = 0.007) (Fig 1B). Group 1 achieved greater weight loss at 1 month in terms of %EWL and %TWL if calculated using each patient's weight at the last clinic visit (Fig 2). There was no difference in terms of %EWL afterward or %TWL between the groups at other time points up to 24 months.

At 12 months, 10 of 12 patients in Group 1 achieved complete diabetes mellitus remission, while all six patients in Group 2 with available data reached complete remission. At one-year, complete remission of hypertension was noted for eight of 14 patients in Group 1 and five of nine patients in Group 2. Among the available lipid profile data, remission was noted in 13 of 15 patients in Group 1 and in 7 of 9 patients in Group 2. There was no difference regarding the comorbidity remission rate between the groups.

## Discussion

Routine intervention via an orlistat-based regimen achieved only modest weight loss, with two-thirds of patients (35/55, 64%) showing a median 1.7% TWL and 3.1% EWL. As evidence

**Table 1. Clinical characteristics of the patients.**

| Variables | Group 1 (n = 35) | Group 2 (n = 20) | P value |
|---|---|---|---|
| Age (years), median (IQR) | 33.0 (44.5–27.5) | 37.5 (48.0–33.0) | 0.180 |
| Female, n (%) | 19 (54%) | 5 (25%) | 0.068 |
| **Weight at last clinic visit** | | | |
| BW (kg), median (IQR) | 134.0 (145.5–123.5) | 132.3 (149.6–123.3) | 0.895 |
| BMI (kg/m$^2$), median (IQR) | 48.9 (52.7–45.8) | 44.6 (46.8–42.8) | 0.003* |
| BMI range (kg/m$^2$), n | | | |
| 42.5–49.9 | 22 | 16 | 0.467 |
| 50–59.9 | 10 | 3 | |
| $\geq$ 60 | 3 | 1 | |
| **Weight at surgery** | | | |
| BW (kg), median (IQR) | 132.6 (144.3–121.5) | 134.7 (152.8–124.8) | 0.306 |
| BMI (kg/m$^2$), median (IQR) | 47.9 (51.5–44.7) | 45.6 (48.0–44.1) | 0.182 |
| BW change (kg), median (IQR) | 2.0 (3.2–1.0) | +2.9 (+2.0 –+5.7) | |
| %EWL, median (IQR) | 3.1% (5.1%–1.4%) | +4.9% (+3.0%–+8.2%) | |
| %TWL, median (IQR) | 1.7% (2.3%–0.7%) | +2.2% (+1.4%–+4.7%) | |
| **Comorbidity, n (%)** | | | |
| Diabetes mellitus | 12 (34%) | 7 (35%) | 1.00 |
| Insulin | 0 | 2 | 0.122 |
| OHA | 12 | 5 | |
| Hypertension | 17 (49%) | 12 (60%) | 0.575 |
| Dyslipidemia | 20 (57%) | 12 (60%) | 1.00 |

*BW*, body weight; *BMI*, body mass index; *%EWL*, percentage excessive weight loss; *%TWL*, percentage total weight loss; *OHA*, oral hypoglycemic agent; *GERD*, gastroesophageal reflux disease.

*$P < 0.05$

points to beneficial effects such as an easier surgical approach [19, 20], the current study confirmed such an advantageous effect as a marked reduction in operation time, although weight loss results at 2 years were unaffected.

When determining the most appropriate bariatric procedure for each individual, several factors need to be considered, including age, BMI, comorbidities, psychological preparedness, and general well-being. However, importantly, there is presently a dearth of established criteria to facilitate this decision-making process. To this end, we have implemented a patient-centered approach that is centered on considering the patients' personal values and their preferences to reduce uncertainty and conflict in their decision-making process [21]. Moreover, it is crucial to account for the disadvantages related to different bariatric procedures. For example, when considering gastric banding, it is important to be aware of the possibility of suboptimal weight loss, despite diligent monitoring and proper adjustment of the band's filling pressure

**Table 2. Surgical characteristics and outcomes.**

| Variables | Group 1 | Group 2 | P value |
|---|---|---|---|
| Operative duration (min), median (IQR) | 107.0 (155.5–80.0) | 140.0 (169.5–100.5) | 0.109 |
| LOS (days), median (IQR) | 3 (3–2) | 3 (3–3) | 0.198 |
| Complications within 30 days, n (%) | 2 (5.7%) | 0 (0%) | 0.528 |

*LOS*, length of stay

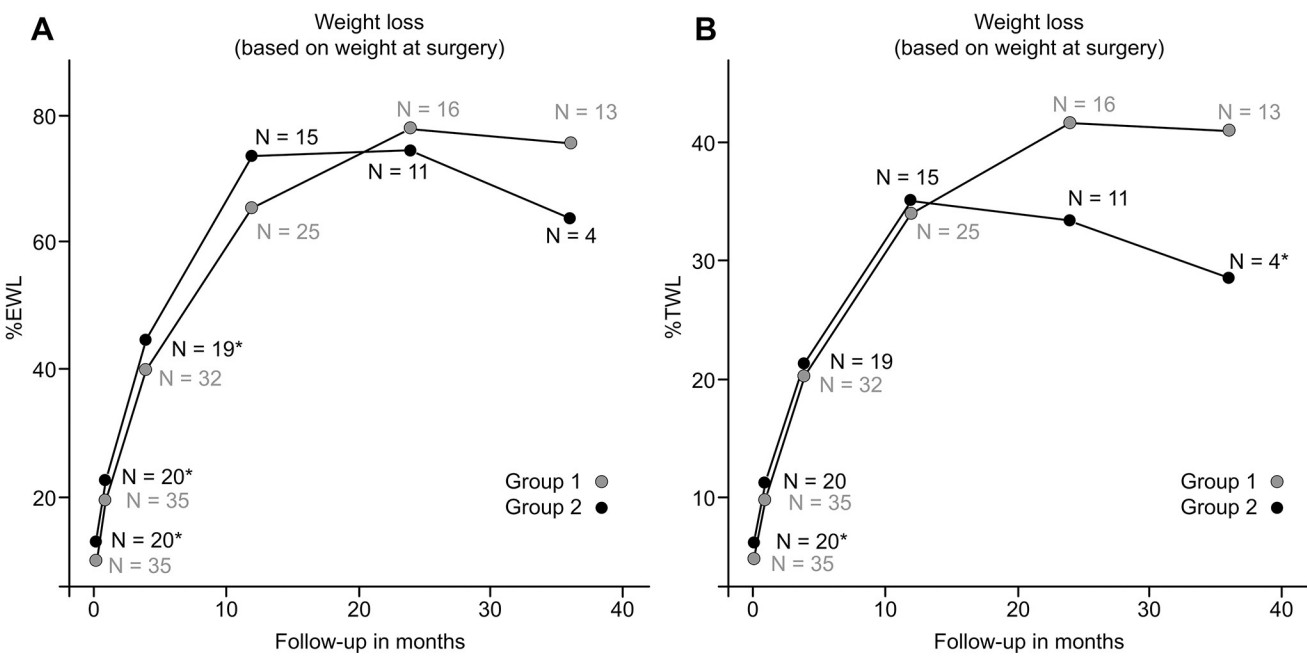

**Fig 1. Weight loss calculated based on weight parameters at surgery.** Group 2 had a significantly greater %EWL than Group 1 up to 4 months postoperatively. The %TWL was significantly greater in Group 2 than in Group 1 at 1 week.

[22]. Although antireflux procedures have been proven to be advantageous after sleeve gastrectomy, there are still lingering concerns about the potential development of GERD and Barrett's esophagus [23]. On the other hand, there are still concerns regarding the potential risks of bile

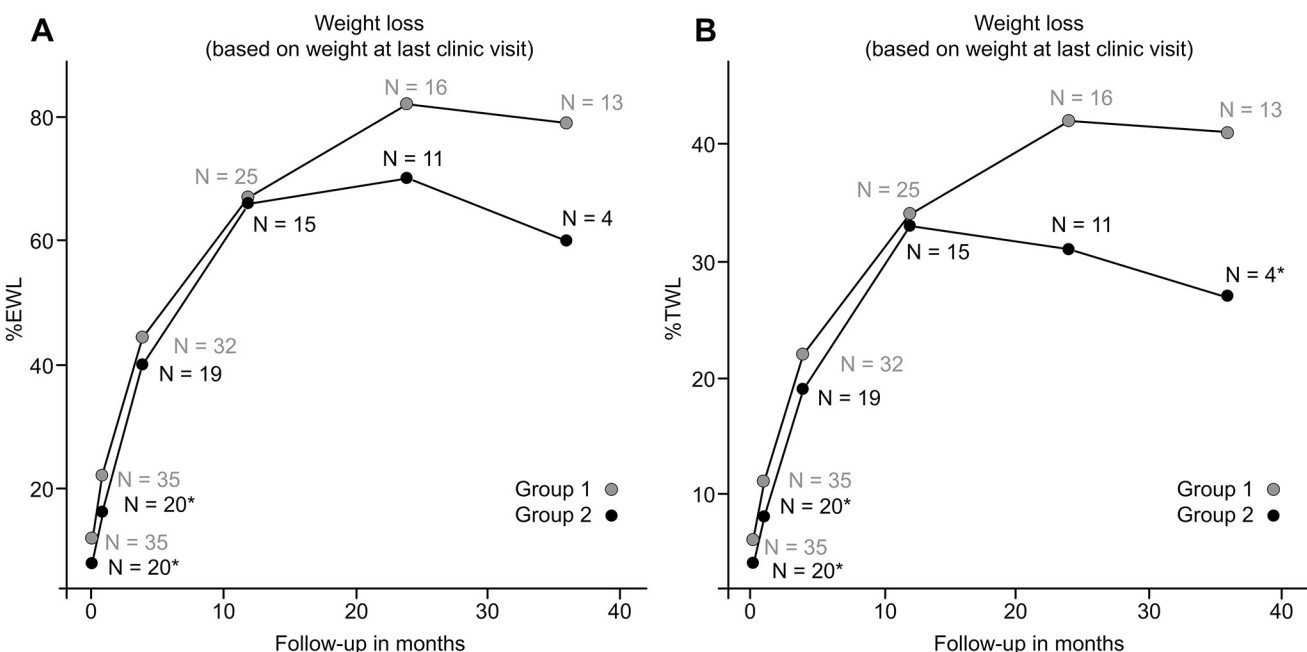

**Fig 2. Weight loss calculated using weight parameters at the last clinical visit.** Weight loss was greater at 1 month in terms of %EWL and %TWL in Group 1.

reflux and adverse events related to poor nutrition following OAGB [24]. Nevertheless, despite current debates and limited long-term data, we have chosen OAGB as our preferred procedure for individuals with severe obesity. This decision is based on its demonstrated effectiveness in terms of weight reduction [13] and the reported safety profile, particularly when compared to Roux-en-Y gastric bypass [16].

The current study did not show causal relationships between orlistat-mediated weight loss and results after OAGB, possibly because gastrointestinal hypoabsorption is only part of a multitude of mechanisms of OAGB. Similarly, regardless of various intervention methods, conflicting results have been reported for subsequent beneficial effects on surgical weight loss [5]. In a randomized controlled trial, Alami et al. [19] found a short-term beneficial effect on weight loss at 3 months after 10% preoperative weight loss. Specifically, Alvarado et al. [20] demonstrated that every 1% increase in preoperative weight loss correlated with a 1.8% increase in EWL at 1 year. In contrast, Horwitz et al. [25] found that insurance-mandated preoperative weight loss was not associated with superior outcomes at 2 years in a retrospective review of 1432 patients. Further conflicting evidence exists, as some studies have reported an inverse relationship between preoperative weight loss and postoperative %EWL [26]. For example, Ochner et al. [1] discovered that presurgical weight gain predicted superior postsurgical weight loss at 3 months. Our research revealed an analogous contradictory finding: Group 2 had a significantly higher %EWL than Group 1 up to 4 months postoperatively (Fig 1A). Potential explanations proposed for this paradoxical finding include increased physiological drive, reduced diet fatigue and an absence of binge episodes postoperatively. Apart from the heterogeneity of study protocols and the relatively small sample sizes of many studies contributing to the aforementioned conflicting results [27], other causes include a lack of unified reporting criteria or methods of quantification. For example, Harnisch et al. [28] found a transient additive effect on weight loss postoperatively if the %EWL was calculated from the initial program-entry weight; similarly, we found such an opposing result, with significantly higher %EWL and %TWL in Group 1 at postoperative month 1 by this calculation method (Fig 2) Notably, Group 2 tended to have a lower BMI at program entry, which might have affected the interpretation, as there was a recognized inverse relationship between BMI and % EWL [26]. However, there is a lack of standardized criteria, and in most studies, the researchers chose to assess the patients' weight loss based on their preoperative weights rather than utilizing their baseline weights at the initiation of the intervention [5];; in view of this, since the BMI difference was not significant at surgery (45.6 kg/m2 vs. 47.9; p = 0.182), the comparison of postoperative weight loss outcomes between different groups remains valid and unaffected by the disparity in BMI at the initiation of the program. In addition, considering that %TWL was less affected in this aspect [26], we also found that the difference between groups was less significant in terms of postoperative %TWL (Fig 1B). Interestingly, since it is not uncommon for patients to have a "last meal" emotional response [3] and exploit final opportunities to overindulge [1], proper weight management has been postulated to halt the natural trend and theoretically bring more successful and sustained weight loss over time for those motivated and with better compliance [29]. In this regard, a more favorable response after intervention in group 1 may reflect a higher motivation and better compliance. In fact, group 1 had a higher follow-up rate and tended to have more %TWL at the 3rd year than group 2 (40% vs. 28%, p = 0.04), which is concordant with the important notion that patients who are compliant with regular follow-up tend to have more successful long-term weight loss [29]. Given the nonnegligibly high attrition rate, small sample size and paucity of timely feedback, it remains to be seen whether a more comprehensive preoperative program effectively increases patients' understanding, promotes healthy behaviors and thereby enhances treatment results.

Notably, prior studies reported significant positive impacts only after substantial weight loss (e.g., > 10% EWL) was achieved [19]. For the same reason [30], along with the limitations of a relatively low frequency of complications and the size of the study groups, the current study did not show advantages in terms of 30-day complications or comorbidity remission. Therefore, the intervention method may be insufficient and lead to a flawed conclusion as an intrinsic weakness of the study design. However, as increasing energy restriction progressively decreases compliance [14], it is common practice to assign patients a self-monitored 4- to 6-week diet regimen [27]. When self-administered orlistat is added for 2 weeks preceding surgery as a final enhancement, the attempt is to capitalize on its potential additive effect, as the decrease in liver volume is reported to occur mostly within the first 2 weeks of weight change [31], and slight short-term weight loss is reportedly associated with a reduction in visceral adiposity, which decreases liver volume and eases the surgical approach [19, 20]. As a result, the current study achieved a high adherence rate and with clinically relevant advantages in terms of reduction in operation time that is in line with the aforementioned evidence and systematic review [4]. Part of the reason may be that our patients have a lower BMI; therefore, weight reduction of this magnitude can still be helpful. The flaw resides in the lack of objective data such as surgeon-reported visibility scores or ultrasound to measure liver volume and support the assumption that it facilitates the surgical approach. More data collection, such as tracking changes in liver function tests, may provide further insight into this aspect.

Notably, the results after preoperative interventions can be unpredictable, whereas some studies reported that patients with a higher initial weight tended to be more successful [32], other determining factors remain elusive. Recognized with the limited efficacy of the current method, the initial encouraging result motivates us to find a more patient-centered preemptive modification, such as conducting a more extended project or utilizing other new listing available medications, such as glucagon-like peptide 1 agonist, for those who appeared to be at risk of failure and capitalized on the beneficial effects.

In summary, short-term compliance with a weight management program facilitates bariatric surgery but does not translate to superior postoperative weight loss results at 2 years.

## Limitations

Owing to the retrospective nature of this study, selection bias could not be completely eliminated despite patients in both groups receiving the same counseling process and postoperative care. Other factors, such as adherence to diet and physical activity, were not directly monitored. Additionally, self-reported compliance can be prone to recall error. Altogether, these pitfalls may influence the research results and cause bias. The strengths of our research were that a single surgeon performed a consistent technique, and all treatment and follow-up were conducted under the same criteria; therefore, the impact of heterogeneous practice can be largely avoided.

## Conclusions

Preoperative weight loss control by an orlistat-based regimen was beneficial for two-thirds of susceptible patients but had no effect on weight loss at 2 years after OAGB. A sophisticated, patient-centered approach is required to accomplish more effective weight loss because there are confirmatory benefits, such as a shortened operation time. A more focused follow-up and comprehensive approach may provide more affirmative data and refine our practice.

## Supporting information

**S1 Dataset. Anonymized dataset.**
(XLSX)

## Author Contributions

**Data curation:** Hung-Chieh Lo, Shih-Chang Hsu.

**Formal analysis:** Hung-Chieh Lo, Shih-Chang Hsu.

**Investigation:** Hung-Chieh Lo, Shih-Chang Hsu.

**Methodology:** Hung-Chieh Lo, Shih-Chang Hsu.

**Supervision:** Hung-Chieh Lo.

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
