## [Decision Letter · Decision Letter 0]

31 May 2023

PONE-D-23-06504Effectiveness of a preoperative orlistat-based weight management plan and its impact on the results of one-anastomosis gastric bypass: A retrospective studyPLOS ONE

Dear Dr. Lo,

Thank you for submitting your manuscript to PLOS ONE. After careful consideration, we feel that it has merit but does not fully meet PLOS ONE’s publication criteria as it currently stands. Therefore, we invite you to submit a revised version of the manuscript that addresses the points raised during the review process.

Please revise.

We look forward to receiving your revised manuscript.

Kind regards,

Academic Editor

PLOS ONE

Journal Requirements:

Reviewers' comments:

Reviewer's Responses to Questions

**Comments to the Author**

1. Is the manuscript technically sound, and do the data support the conclusions?

Reviewer #1: Yes

Reviewer #2: Partly

2. Has the statistical analysis been performed appropriately and rigorously? 

Reviewer #1: Yes

Reviewer #2: No

3. Have the authors made all data underlying the findings in their manuscript fully available?

Reviewer #1: Yes

Reviewer #2: No

4. Is the manuscript presented in an intelligible fashion and written in standard English?

Reviewer #1: Yes

Reviewer #2: Yes

5. Review Comments to the Author

Reviewer #1: In this paper the Authors aim to assess the effectiveness of a preoperative orlistat-based weight management plan and its impact on the results of one-anastomosis gastric bypass. It is an interesting and debated topic. A comprehensive and extensive literature review of the NCBI database PubMed was also carried out. The article was well conducted and it is interesting in its fields. It is a well-structured paper, written in good English and the References are up dated.

Minor issues:

In the “discussion” section I suggest to better analyze the drawbacks of the different bariatric procedures. Therefore, the following paper should be considered:

“Lucido FS, Scognamiglio G, Nesta G, Del Genio G, Cristiano S, Pizza F, Tolone S, Brusciano L, Parisi S, Pagnotta S, Gambardella C. It is really time to retire laparoscopic gastric banding? Positive outcomes after long-term follow-up: the management is the key. Updates Surg. 2022 Apr;74(2):715-726. doi: 10.1007/s13304-021-01178-1. Epub 2021 Oct 1. PMID: 34599469; PMCID: PMC8995288.”

“Del Genio G, Tolone S, Gambardella C, Brusciano L, Volpe ML, Gualtieri G, Del Genio F, Docimo L. Sleeve Gastrectomy and Anterior Fundoplication (D-SLEEVE) Prevents Gastroesophageal Reflux in Symptomatic GERD. Obes Surg. 2020 May;30(5):1642-1652. doi: 10.1007/s11695-020-04427-1. PMID: 32146568.”

“Pizza F, Lucido FS, D'Antonio D, Tolone S, Gambardella C, Dell'Isola C, Docimo L, Marvaso A. Biliopancreatic Limb Length in One Anastomosis Gastric Bypass: Which Is the Best? Obes Surg. 2020 Oct;30(10):3685-3694. doi: 10.1007/s11695-020-04687-x. PMID: 32458362.”

Reviewer #2: Dear authors, I read our manuscript carefully. I have few comments and questions and hope addressing them can robust the quality of your upcoming paper.

1- Why did you set the BMI ≥42.5 kg/m2 to prescribe Orlistat? Did you have any reference or guideline to recommend it?

2- Did you use a constant 200-cm biliopancreatic limb in all patients irrespective of age, sex and BMI?

3- Why did you calculate excessive weight loss (%EWL) using the ideal body weight equivalent to a BMI of 23.5 kg/m2 instead of 25 kg/m2?

4- Did you have a selection criteria for OAGB in addition to BMI>50? I recommend to use the paper "Patient Selection in One Anastomosis/Mini Gastric Bypass-an Expert Modified Delphi Consensus." (PMID: 35704259 DOI: 10.1007/s11695-022-06124-7) to respond it.

5- In table-1, you showed a statistically significant difference in pre-op BMI between two groups? How did you eliminate these effect (non-matched groups) at final conclusion?

6- Please clarify that how did you evaluate the patients' adherence to use orlistat before the surgery? Did you have a questionnaire?

6. PLOS authors have the option to publish the peer review history of their article (what does this mean?). If published, this will include your full peer review and any attached files.

Reviewer #1: No

Reviewer #2: **Yes: **Mohammad Kermansaravi

---

## [Author Response · Author response to Decision Letter 0]

16 Jun 2023

(reviewer #1, general comment)

In this paper the Authors aim to assess the effectiveness of a preoperative orlistat-based weight management plan and its impact on the results of one-anastomosis gastric bypass. It is an interesting and debated topic. A comprehensive and extensive literature review of the NCBI database PubMed was also carried out. The article was well conducted and it is interesting in its fields. It is a well-structured paper, written in good English and the References are up dated.

Response:

We genuinely value the editor's overall feedback regarding this article.

(reviewer #1, comment #2)

In the “discussion” section I suggest to better analyze the drawbacks of the different bariatric procedures. Therefore, the following paper should be considered:

“Lucido FS, Scognamiglio G, Nesta G, Del Genio G, Cristiano S, Pizza F, Tolone S, Brusciano L, Parisi S, Pagnotta S, Gambardella C. It is really time to retire laparoscopic gastric banding? Positive outcomes after long-term follow-up: the management is the key. Updates Surg. 2022 Apr;74(2):715-726. doi: 10.1007/s13304-021-01178-1. Epub 2021 Oct 1. PMID: 34599469; PMCID: PMC8995288.”

“Del Genio G, Tolone S, Gambardella C, Brusciano L, Volpe ML, Gualtieri G, Del Genio F, Docimo L. Sleeve Gastrectomy and Anterior Fundoplication (D-SLEEVE) Prevents Gastroesophageal Reflux in Symptomatic GERD. Obes Surg. 2020 May;30(5):1642-1652. doi: 10.1007/s11695-020-04427-1. PMID: 32146568.”

“Pizza F, Lucido FS, D'Antonio D, Tolone S, Gambardella C, Dell'Isola C, Docimo L, Marvaso A. Biliopancreatic Limb Length in One Anastomosis Gastric Bypass: Which Is the Best? Obes Surg. 2020 Oct;30(10):3685-3694. doi: 10.1007/s11695-020-04687-x. PMID: 32458362.”

Response:

We express our gratitude to the reviewer for bringing attention to this significant oversight in the manuscript and for providing the corresponding references. In response, we have made revisions to the Discussion section to emphasize the importance of the procedure selection process and to outline the drawbacks associated with various bariatric procedures.

Page 13-14, Discussion, Line 214-229 now reads as follows:

When determining the most appropriate bariatric procedure for each individual, several factors need to be considered, including age, BMI, comorbidities, psychological preparedness, and general well-being. However, importantly, there is presently a dearth of established criteria to facilitate this decision-making process. To this end, we have implemented a patient-centered approach that is centered on considering the patients’ personal values and their preferences to reduce uncertainty and conflict in their decision-making process [22]. Moreover, it is crucial to account for the disadvantages related to different bariatric procedures. For example, when considering gastric banding, it is important to be aware of the possibility of suboptimal weight loss, despite diligent monitoring and proper adjustment of the band's filling pressure [23]. Although antireflux procedures have been proven to be advantageous after sleeve gastrectomy, there are still lingering concerns about the potential development of GERD and Barrett's esophagus [24]. On the other hand, there are still concerns regarding the potential risks of bile reflux and adverse events related to poor nutrition following OAGB [25]. Nevertheless, despite current debates and limited long-term data, we have chosen OAGB as our preferred procedure for individuals with severe obesity. This decision is based on its demonstrated effectiveness in terms of weight reduction [26] and the reported safety profile, particularly when compared to Roux-en-Y gastric bypass [17].

(reviewer #2, comment #1)

Why did you set the BMI ≥42.5 kg/m2 to prescribe Orlistat? Did you have any reference or guideline to recommend it?

Response:

As there are no standardized guidelines, the preoperative determination in the BMI cutoff point for initiating preoperative weight management programs is based on individual assessment and greatly influenced by the specific characteristics of the study populations. For instance, a systematic review highlighted the significant variation, as the baseline BMI ranged from 43.5 to 58.4 kg/m2. In Sweden, a 2- to 3-week low-calorie diet regimen was implemented prior to bariatric surgery, regardless of the preoperative BMI. The mean BMI at surgical planning was approximately 42 kg/m2.

To enhance the clarity of the study's details, we have incorporated the subsequent paragraphs.

Page 4, Introduction, Line 53-60 now reads as follows:

Due to the lack of standardized guidelines, the preoperative determination of body mass index (BMI) cutoff points is based on individual assessment and greatly influenced by the specific characteristics of the study populations. For example, a systematic review revealed substantial variation in the BMI cutoff point for initiating weight management programs, as the baseline BMI ranged from 43.5 to 58.4 kg/m2 [8]. In Sweden, the average BMI at the time of surgical planning was 42 kg/m2, and a 2- to 3-week low-calorie diet regimen was routinely implemented, regardless of the preoperative BMI [9].

Previously, we provided an explanation for establishing the intervention threshold at a lower BMI and we have made revisions to the relevant section as follows:

Page 3-4, Introduction, Line 60-66 now reads as follows:

Studies have shown that in terms of BMI, Asians tend to have a higher body fat percentage than matched Caucasians and a higher prevalence of central adiposity than Europeans [10,11]. We have also observed in our own prior experience that the success of the procedure was hindered in patients with a relatively low BMI (ranging from 40.8 to 48.1 kg/m2) and a voluminous liver [12]; as the goals are to ensure the safety of our patients and the success of the procedure, we decided to implement a prudent preoperative weight management program for patients with a lower BMI than that specified in Western reports [13]. 

(reviewer #2, comment #2)

Did you use a constant 200-cm biliopancreatic limb in all patients irrespective of age, sex and BMI?

Response:

We thank the editor for noting this major omission in our previous writing. The length of the biliary limb and its potential long-term consequences after One Anastomosis Gastric Bypass (OAGB) is a concern in terms of possible causing complications. These complications include bile reflux, nutritional deficiencies, and gallstones. While further research is needed to gain a better understanding of the most suitable length of the biliary limb and its impact on long-term outcomes after OAGB, in our region, a fixed 200-cm biliopancreatic limb length is commonly used due to its reported safety and effectiveness. Therefore, we have chosen to utilize this technique.

The paragraph in question has been modified to provide a more comprehensive explanation for our approach.

Page 8, Materials and methods, lines 132-134 now reads as follows:

Due to its reported safety and efficacy, a biliopancreatic limb with a fixed length of 200 cm was used [17]

(reviewer #2, comment #3)

Why did you calculate excessive weight loss (%EWL) using the ideal body weight equivalent to a BMI of 23.5 kg/m2 instead of 25 kg/m2?

Response:

We sincerely appreciate the reviewer for highlighting the significance of the baseline reference, as it can have a profound impact on the study results. Rather than utilizing a BMI of 25 kg/m2, being overweight was defined as having a BMI greater than 23 according to the Bariatric Society Asia-Pacific Perspective. In Taiwan, being overweight is defined as having a BMI greater than 24. To address the regional criteria mentioned above in our study population, which is composed of individuals from Taiwanese and Asia-Pacific ethnic backgrounds, we set a baseline standard BMI of 23.5. Nevertheless, the study findings remain consistent even when employing a BMI reference of 24.

Acknowledging the significant oversight, relevant text in the Materials and Methods section has been revised to provide further clarification on the reasoning behind the selected baseline BMI.

Page 9, Materials and methods, line 143-146 now read as follows:

A BMI of 23.5 kg/m2 was used when individuals from Taiwanese [18] and Asia-Pacific ethnic backgrounds were included in the study population [19].

(reviewer #2, comment #4)

Did you have a selection criteria for OAGB in addition to BMI>50? I recommend to use the paper "Patient Selection in One Anastomosis/Mini Gastric Bypass-an Expert Modified Delphi Consensus." (PMID: 35704259 DOI: 10.1007/s11695-022-06124-7) to respond it.

Response:

We express our sincere gratitude to the editor for providing their valuable suggestion. We have taken the suggestion into account and included an important reference in the article to enhance the clarity and comprehensiveness of the selection process.

Page 8, Materials and methods, line 127-129 now read as follows:

The current general selection criteria correspond with the previous selection guidelines in that grade C or D gastroesophageal reflux disease (GERD) or Barrett's esophagus are contraindications for the index procedure in both guidelines [16].

(reviewer #2, comment #5)

In table-1, you showed a statistically significant difference in pre-op BMI between two groups? How did you eliminate these effect (non-matched groups) at final conclusion

Response:

We are grateful to the editor for highlighting this important issue that could potentially lead to an erroneous intergroup comparison and impact the study outcome. Specifically, it was noted that Group 2 had a lower BMI at the time of program entry, which could have influenced the interpretation of the results, considering the acknowledged inverse relationship between BMI and %EWL. Given the retrospective nature of the study, the limited sample size, and the study focus on a specific subpopulation, it appears challenging to perform a matched comparison or control for all confounding factors, such as age, sex, and BMI. However, there is a lack of standardized criteria and in most studies, the researchers chose to assess the patients’ weight loss based on their preoperative weights rather than utilizing their baseline weights at the initiation of the intervention, as the BMI difference was not statistically significant at the time of surgery (45.6 kg/m2 vs. 47.9; p=0.182), the comparison of postoperative weight loss outcomes between the different groups remains valid and unaffected by the disparity in BMI at the initiation of the program.

To provide a clearer explanation and to reinforce the rationale behind the intergroup comparison, we have made revisions to the following paragraphs to elucidate these points.

Page 15, Discussion, lines 251-252 now read as follows:

Notably, Group 2 tended to have a lower BMI at program entry

Page 15, Discussion, lines 253-258 now read as follows:

However, there is a lack of standardized criteria, and in most studies, the researchers chose to assess the patients’ weight loss based on their preoperative weights rather than utilizing their baseline weights at the initiation of the intervention [31]; in view of this, since the BMI difference was not significant at surgery (45.6 kg/m2 vs. 47.9; p=0.182), the comparison of postoperative weight loss outcomes between different groups remains valid and unaffected by the disparity in BMI at the initiation of the program.

(reviewer #2, comment #6) 

Please clarify that how did you evaluate the patients' adherence to use orlistat before the surgery? Did you have a questionnaire?

Response:

We sincerely appreciate the editor for bringing this significant omission to our attention. In fact, we are utilizing a comprehensive questionnaire that inquires about dietary modifications and gastrointestinal adverse events. Additionally, we specifically assessed the frequency and timing of daily medication use to ensure the accuracy and reliability of self-reported outcomes and to assess medication adherence.

We have provided an additional description in the Materials and methods section to provide a clearer explanation of the process.

Page 8, Materials and methods, line 134-139 now read as follows:

The responsible staff recorded the patient’s weight control methods by utilizing a comprehensive questionnaire that inquiries about dietary modifications and gastrointestinal adverse events. Additionally, we specifically assessed the frequency and timing of daily medication use to ensure the accuracy and reliability of self-reported outcomes and to assess medication adherence.

The limitations of this method have been highlighted in the corresponding section of the previous manuscript.

Page 17, Limitations, line 305 read as follows:

Self-reported compliance can be prone to recall error.

---

## [Decision Letter · Decision Letter 1]

10 Jul 2023

Effectiveness of a preoperative orlistat-based weight management plan and its impact on the results of one-anastomosis gastric bypass: A retrospective study

PONE-D-23-06504R1

Dear Dr. Lo,

We’re pleased to inform you that your manuscript has been judged scientifically suitable for publication and will be formally accepted for publication once it meets all outstanding technical requirements.

Kind regards,

Academic Editor

PLOS ONE

Additional Editor Comments (optional):

Reviewers' comments:

Reviewer's Responses to Questions

**Comments to the Author**

1. If the authors have adequately addressed your comments raised in a previous round of review and you feel that this manuscript is now acceptable for publication, you may indicate that here to bypass the “Comments to the Author” section, enter your conflict of interest statement in the “Confidential to Editor” section, and submit your "Accept" recommendation.

Reviewer #1: All comments have been addressed

Reviewer #2: All comments have been addressed

2. Is the manuscript technically sound, and do the data support the conclusions?

Reviewer #1: Yes

Reviewer #2: Yes

3. Has the statistical analysis been performed appropriately and rigorously? 

Reviewer #1: (No Response)

Reviewer #2: Yes

4. Have the authors made all data underlying the findings in their manuscript fully available?

Reviewer #1: (No Response)

Reviewer #2: Yes

5. Is the manuscript presented in an intelligible fashion and written in standard English?

Reviewer #1: Yes

Reviewer #2: Yes

6. Review Comments to the Author

Reviewer #1: Thank you for accepting my suggestions. In my opinion it is suitable for publication in its present form

Reviewer #2: Congratulations to the authors for their new edition of manuscript.

All my comments are addressed and I think this version can be acceptable for publication.

7. PLOS authors have the option to publish the peer review history of their article (what does this mean?). If published, this will include your full peer review and any attached files.

Reviewer #1: **Yes: **Claudio Gambardella

Reviewer #2: **Yes: **Mohammad Kermansaravi

---

## [Editor Report · Acceptance letter]

20 Jul 2023

PONE-D-23-06504R1 

Effectiveness of a preoperative orlistat-based weight management plan and its impact on the results of one-anastomosis gastric bypass: A retrospective study 

Dear Dr. Lo:

I'm pleased to inform you that your manuscript has been deemed suitable for publication in PLOS ONE. Congratulations! Your manuscript is now with our production department. 

Kind regards, 

on behalf of

Dr. Robert Jeenchen Chen 

Academic Editor

PLOS ONE